# Chrysin Induces Apoptosis and Autophagy in Human Melanoma Cells via the mTOR/S6K Pathway

**DOI:** 10.3390/biomedicines10071467

**Published:** 2022-06-21

**Authors:** Jae-Han Lee, Eun-Seon Yoo, So-Hee Han, Gi-Hwan Jung, Eun-Ji Han, Eun-Young Choi, Su-ji Jeon, Soo-Hyun Jung, Bumseok Kim, Sung-Dae Cho, Jeong-Seok Nam, Changsun Choi, Jeong-Hwan Che, Ji-Youn Jung

**Affiliations:** 1Department of Companion, Laboratory Animal Science, Kongju National University, Yesan 32439, Korea; tacop1013@gmail.com (J.-H.L.); aileen_00@naver.com (E.-S.Y.); crong524@naver.com (S.-H.H.); ghjung4491@gmail.com (G.-H.J.); hansuran9101@naver.com (E.-J.H.); secure801@gmail.com (E.-Y.C.); jsz0223@naver.com (S.-j.J.); jsh0019y@gmail.com (S.-H.J.); 2College of Veterinary Medicine, Bio-Safety Research Institute, Jeonbuk National University, Iksan 54896, Korea; bskims@jbnu.ac.kr; 3Department of Oral Pathology, School of Dentistry, Dental Research Institute, Seoul National University, Seoul 03080, Korea; efiwdsc@snu.ac.kr; 4Gwangju Institute of Science and Technology, School of Life Sciences, Gwangju 61005, Korea; namje@gist.ac.kr; 5School of Food Science and Technology, Chung-ang University, Ansung 17456, Korea; cchoi@cau.ac.kr; 6Biomedical Center for Animal Resource Development, Seoul National University College of Medicine, Seoul 03080, Korea; casache@snu.ac.kr; 7Biomedical Research Institute, Seoul National University Hospital, Seoul 03080, Korea

**Keywords:** chrysin, chrysin-induced autophagy, melanoma, chemotherapy, apoptosis

## Abstract

Chrysin is known to exert anti-inflammatory, antioxidant, and anticancer effects. The aim of this study was to investigate the anticancer effects of chrysin in the human melanoma cells A375SM and A375P. The results obtained demonstrated successful inhibition of the viability of these cells by inducing apoptosis and autophagy. This was confirmed by the level of apoptosis-related proteins: Bax and cleaved poly (ADP-ribose) polymerase both increased, and Bcl-2 decreased. Moreover, levels of LC3 and Beclin 1, both autophagy-related proteins, increased in chrysin-treated cells. Autophagic vacuoles and acidic vesicular organelles were observed in both cell lines treated with chrysin. Both cell lines showed different tendencies during chrysin-induced autophagy inhibition, indicating that autophagy has different effects depending on the cell type. In A375SM, the early autophagy inhibitor 3-methyladenine (3-MA) was unaffected; however, cell viability decreased when treated with the late autophagy inhibitor hydroxychloroquine (HCQ). In contrast, HCQ was unaffected in A375P; however, cell viability increased when treated with 3-MA. Chrysin also decreased the phosphorylation of mTOR/S6K pathway proteins, indicating that this pathway is involved in chrysin-induced apoptosis and autophagy for A375SM and A375P. However, studies to elucidate the mechanisms of autophagy and the action of chrysin in vivo are still needed.

## 1. Introduction

The incidence of human melanoma, the most malignant form of skin cancer, has been increasing in recent years [1]. Moreover, melanoma has a high metastatic rate. It is also highly resistant to radiation therapy and chemotherapy. Therefore, early detection combined with surgical resection is the only treatment for melanoma [2,3,4]. In the West, melanoma is being actively studied because of its high incidence and mortality rates. However, the related research in Korea is limited [5,6].

Chrysin is a natural flavonoid found in honey, propolis, and passionflower [7,8]. Its physiological effects include anti-inflammatory, anti-oxidant, and anticancer effects [9]. The anticancer effect of chrysin has been confirmed in HeLa cells for spinal cancer and in U937 and HL60 cells for leukemia [10]. Chrysin has also been confirmed to be effective in melanoma cells such as B16-F1 and A375 cells [11]. However, studies on A375SM, with high invasive and metastatic properties, and A375P, with low invasive and metastatic properties, have not been conducted [12].

Apoptosis is a defense mechanism that eliminates cells with intracellular DNA damage or viral infection. It is being studied as an action mechanism of chemotherapy to inhibit the progression of cancer [13]. Apoptosis is controlled largely by the Bcl-2 family, which is divided into pro-apoptotic and anti-apoptotic proteins [14]. Pro-apoptotic proteins include Bax, Bid, and Bad. Anti-apoptotic proteins include Bcl-2, Bcl-xL, and A1 [15,16,17].

Autophagy is a physiological process that affects several functions such as homeostasis maintenance, developmental processes, and immune functions. Among autophagy proteins, autophagy-related protein and protein light chain 3 (LC3) play an important role in the initial formation of autophagosomes. Beclin 1 is involved in the formation of autophagosomes and facilitates autophagy proteins in the cytoplasm [18]. Autophagy is achieved by enclosing unnecessary intracellular organelles in autophagosomes and destroying them by binding to lysosomes [19]. It helps cells survive various stress conditions such as lack of nutrients and infection by pathogens. Therefore, autophagy protects cells against substances such as anticancer drugs. However, the autophagy and apoptosis systems are interconnected, and they might not necessarily be involved in these tasks [20,21,22]. Therefore, the regulation of autophagy is expected to affect the occurrence and treatment of various diseases, including cancer [23,24].

mTOR is a catalytic component of two protein complexes, mTORC1 and mTORC2. Akt activates mTORC1 by phosphorylating and inhibiting TSC1/2 and phosphorylates and activates mTORC2. Therefore, mTOR functions both upstream and downstream of Akt [25]. Abnormal expression of Akt/mTOR frequently occurs in the pathogenesis of cancer. Therefore, this route is substantially related to the prevention and treatment of cancer [26,27,28]. Studies have reported that the mTOR pathway affects autophagy, apoptosis, and cell proliferation [29,30,31]. S6 kinase, a downstream factor of mTOR, promotes cell proliferation and migration in response to signaling of the AMPK/mTOR/S6K1 pathway [32].

In this study, we investigated the anticancer effects of chrysin in human melanoma cell lines A375SM and A375P and attempted to elucidate the mechanism underlying its effects. Moreover, the relationship between chrysin-induced apoptosis and autophagy in the melanoma cells was studied.

## 2. Material and Methods

### 2.1. Reagents and Antibodies

DMEM was purchased from Welgene (Gyeonsan, Korea). Fetal bovine serum (FBS) and penicillin were purchased from Gibco BRL (Grand Island, NY, USA). Common reagents, chrysin (Figure 1A), 3-(4,5-dimethylthiazol-2-yl)-2,5-diphenyltetrazolium bromide (MTT), and 4′,6-diamidino-2-phenylindole (DAPI) were purchased from Sigma-Aldrich Co. (St. Louis, MO, USA). The FITC-annexin-V detection kit was purchased from BD PharmingenTM (San Diego, CA, USA). Anti-Bax, anti-Bcl-2, anti-poly ADB ribose polymerase (PARP), anti-Beclin 1, anti-LC3, and anti-IgG primary antibodies were purchased from Cell Signaling (Beverly, MA, USA). Anti-β actin and anti-IgG were purchased from Santa Cruz biotechnology Inc. (St. Finnell, Dallas, USA). The early inhibitor of autophagy 3-methyladenine (3-MA) and the late inhibitor hydroxychloroquine (HCQ) were purchased from Selleck Chemical (Houston, TX, USA).

### 2.2. Cell Culture

A375SM and A375P were purchased from Korea Cell Line Bank. They were cultured using DMEM containing 5% FBS and 1% penicillin/streptomycin at 37 °C under 5% CO_2_. 

### 2.3. MTT Assay

The MTT assay was performed to confirm that chrysin inhibits the viability of melanoma cells. A375SM and A375P were plated in a 96-well plate at 2 × 10⁴ cells/mL and were incubated for 24 h. They were then treated with chrysin at concentrations of 0, 20, 40, 60, 80, and 100 µM. The samples in the 96-well plate were treated with chrysin for 24 h. Thereafter, 40 µL of MTT (1 mg/mL) was added, and the samples were incubated at 37 °C under 5% CO_2_ for 2 h. Following incubation, the MTT reagent was removed and the cells were treated with DMSO at 100 µL/well to dissolve all formazan formed in the wells. The absorbance of the samples was measured at 595 nm using an ELISA-reader (Bio-Rad Laboratories Inc., Hercules, CA, USA).

### 2.4. DAPI Staining

DAPI staining was performed to observe the specific nuclear morphological changes that occurred when apoptosis was induced. A375SM and A375P cells were plated in a 60 mm dish at 1 × 10^5^ cells/mL and incubated for 24 h. The cells were treated with chrysin at concentrations of 0, 40, and 80 µM and were incubated for 24 h. The cells were then washed with phosphate-buffered saline (PBS) and fixed with 4% formaldehyde for 15 min. Following another PBS wash, 2 mL of DAPI reagent was added, and the cells were observed at 200× using a fluorescence microscope (Zeiss Fluorescence Microscope, Thornwood, NY, USA). Quantification is expressed as a percentage after counting DAPI-positive cells in one screen. 

### 2.5. Acridine Orange Staining

Acridine orange was used to observe acidic vesicular organelles, one of the morphological features of autophagy. The cells were seeded in a 60-mm dish at 2 × 10^5^ cells/mL, incubated for 24 h, treated with chrysin at concentrations of 0, 40, and 80 µM, and incubated again for 24 h in a CO_2_ incubator. After 24 h, the medium containing chrysin was removed, and the cells were washed twice with PBS. After washing, the cells were fixed with 4% paraformaldehyde for 15 min. The cells were then washed with PBS to remove the paraformaldehyde, treated with 2 mL of acridine orange solution (5 µg/mL), and incubated for 10 min for the reaction to occur. Thereafter, the cells were observed under a fluorescence microscope.

### 2.6. Flow Cytometric Analysis 

Annexin-V and propidium iodide staining was performed to analyze the degree of apoptosis induced by chrysin in melanoma cells quantitatively. A375SM and A375P cells were treated with chrysin at concentrations of 0, 40, and 80 µM and incubated for 24 h. The cells were detached using trypsin–EDTA and centrifuged to obtain a cell pellet. The cell pellet was washed with PBS and centrifuged again. The pellet was re-suspended in a 1× binding buffer at 2 × 10^5^ cells/mL. Annexin-V and propidium iodide were added to the cells, which were then incubated for 15 min and measured using a FACSCalibur™ flow cytometer (BD Biosciences, NJ, USA). 

### 2.7. Western Blotting

Western blotting was performed to confirm the expression of apoptosis-related proteins. A375SM and A375P cells were cultured in a 75T Flask at 37 °C using 5% CO_2_ for 24 h. They were treated with chrysin at concentrations of 0, 40, and 80 μM. The cells were detached using trypsin–EDTA and centrifuged. They were washed with PBS and centrifuged again; cell lysis buffer (Invitrogen, Carlsbad, CA, USA) was added to the pellet, which was incubated at 4 °C for 20 min. The lysate was centrifuged, and the supernatant was used as the cell lysate. The concentration of the extracted protein was measured using the Bradford protein assay. The extracted protein was separated by electrophoresis using a 12% sodium dodecyl sulfate-polyacrylamide gel, and it was then transferred to a nitrocellulose membrane. Blocking was performed by treating the protein-transferred membrane with 5% skim milk for 1 h and 30 min. After blocking, primary antibodies were added, and the membrane was incubated overnight at 4 °C. Subsequently, anti-rabbit IgG and anti-mouse IgG were added, and the membrane was incubated for 2 h. Protein bands were confirmed using ECL-detection reagents (Pierce, Rockford, IL, USA). The density of each band was measured using the image program Image J Launcher (provided by NCBI).

### 2.8. Statistical Analysis

All experimental results are expressed as mean and standard deviation. For comparison between groups, a one-way ANOVA was performed, followed by Student’s *t*-test. The difference was determined to be statistically significant when *p* < 0.05, compared with the control group.

## 3. Results

### 3.1. Effects of Chrysin on A375SM and A375P Cell Viability

To confirm the effects of chrysin on the viability of A375SM and A375P melanoma cells, they were treated with chrysin for 24 h at different concentrations (0, 20, 40, 60, 80, and 100 µM) and analyzed using the MTT assay. After 24 h of treatment, the viability of the cells was confirmed to decrease with increasing chrysin concentration. In A375SM, the viability decreased to 96.1%, 83.3%, 67.6%, 58.6%, and 41.1%, and in A375P, it decreased to 96.8%, 76.6%, 62.3%, 50.9%, and 34.9% as chrysin concentration increased (Figure 1B). These inhibitory effects showed statistical significance from 40 µM in both cell lines.

### 3.2. Chrysin-Induced Apoptosis

To determine whether apoptosis had caused the reduction in the viability of A375SM and A375P cells after chrysin treatment, the changes in nuclear morphology were observed through DAPI staining. As shown in Figure 2A, no noticeable change was observed in the control group; however, in the chrysin-treated group, nuclear and chromatin condensation was observed in both A375SM and A375P. There was a particularly significant difference at 100 µM (Figure 2A). The number of DAPI-positive cells at 0, 40, and 80 µM increased to 5.24%, 10.02%, and 20.02% in A375SM and 2.37%, 8.86%, and 26.83% in A375P, respectively (Figure 2B). Moreover, early/late apoptosis was observed using flow cytometric analysis with annexin V and propidium iodide staining to confirm the degree of cell death. The results revealed that early apoptosis and late apoptosis increased with the concentration of chrysin in both A375SM and A375P cells (Figure 3A). Total apoptosis, including early and late apoptosis, increased in a concentration-dependent manner compared with that in the control group, to 19.32%, 28.19%, and 39.71% in A375SM and 18.62%, 29.58%, and 58.85% in A375P, respectively (Figure 3B). These results confirmed that apoptosis inhibited the proliferation of A375SM and A375P cells. Moreover, the results of western blot confirmed that the expression of Bax, a protein known to induce apoptosis, increased and the expression of bcl-2, a protein known to suppress apoptosis, decreased with increasing concentration in both A375SM and A375P. Fragmentation of PARP, which is involved in DNA repair, and caspase substrate were observed (Figure 4).

### 3.3. Chrysin-Induced Autophagy

After chrysin treatment, A375SM and A375P cells showed different morphological changes, such as vacuole formation, compared with those in the control group (Figure 5A). To determine whether these morphological changes were caused by autophagy, the expression level of the acidic vesicular organelles (autolysosome) was assessed. The results of acridine orange staining, which stains acidic vesicular organelles, revealed that both melanoma cell lines only showed slight changes in the control group; however, the number of positive cells increased as the concentration increased (Figure 5B). Western blotting was also performed to confirm the expression levels of the autophagy-related proteins. The results confirmed that the expression of Beclin 1 increased relative to the formation of autophagosomes. Moreover, the modification of the microtubule-associated protein, a light chain 3 (LC3) protein, is required to form a double-membrane autophagosome; therefore, the expression of LC3-Ⅰ and LC-3Ⅱ also increased (Figure 6). This confirmed that chrysin induces autophagy in the A375SM and A375P cell lines.

### 3.4. Induction and Inhibition of Apoptosis through Autophagy Inhibition 

To confirm the induction and suppression of apoptosis induced through autophagy inhibition, 3-methyladenine (3-MA, early-stage inhibitor of autophagy) and hydroxychloroquine (HCQ, late-stage inhibitor of autophagy) were used to examine cell viability and protein patterns, respectively. In A375SM, when autophagy was inhibited using 3-MA, there was no difference compared with cells subjected to chrysin treatment alone. In terms of inhibition after treatment with HCQ, cell viability decreased compared to that in cells subjected to chrysin treatment alone. In A375P, cell viability increased with inhibition after treatment with 3-MA compared to that in cells subjected to chrysin treatment alone, and no significant difference was observed when treated with HCQ (Figure 7A). The expression levels of the proteins were determined using Western blotting. When late-stage autophagy was suppressed with HCQ in A375SM cells, Bax increased and Bcl-2 decreased compared with their levels in cells subjected to chrysin treatment alone. In contrast, when autophagy was inhibited in an early stage with 3-MA in A375P, Bax decreased and Bcl-2 increased compared with their levels in cells subjected to chrysin treatment alone (Figure 7B). Therefore, autophagy protects A375SM cells and induces cell death in A375P cells, suggesting that the same compound can affect the inhibition period. The results also suggest that the role of autophagy varies with the cell type.

### 3.5. Induction of Apoptosis and Autophagy via the mTOR Pathway

The mTOR pathway has ribosomal S6 kinase (S6K), ribosomal protein S6 kinase beta-1 (p70S6K), and 4E binding protein 1 (4eBP1) as its downstream factors. The inhibition of the mTOR pathway is associated with the induction of apoptosis and autophagy [27,33]. Therefore, the expression of proteins related to the mTOR pathway was examined using Western blotting. Both A375SM and A375p cells showed a decrease in the expression of p-mTOR, p-P70S6K, p-S6K, and p-4EBP1 with increasing concentrations of chrysin. Therefore, the chrysin-induced apoptosis and autophagy might be related to mTOR (Figure 8). 

## 4. Discussion

In this study, the anticancer effects of chrysin in the melanoma cell lines A375SM and A375P were confirmed to be due to apoptosis and autophagy. Moreover, we revealed the relationship between apoptosis and autophagy. To assess the viability of the melanoma cells after chrysin treatment, A375SM and A375P cells were treated with chrysin at various concentrations that did not affect normal cells (20, 40, 60, and 80 µM) [34] for 24 h, and cell viability was confirmed using an MTT assay. Both cells showed a significant decrease in viability starting from 40 µM (Figure 1). DAPI staining and annexin-V-propidium iodide staining were performed to determine whether the decrease in cell viability was caused by apoptosis. DAPI attaches to divided chromosomes and stains a specific structure called an apoptotic body [35]. According to a previous study in the melanoma cells B16F1 and A375 [11], DAPI-positive cells were observed after chrysin treatment. Similarly, in this study, both A375SM and A375P cells showed an increase in DAPI-positive cells after chrysin treatment, and the number tended to increase with the increase in concentration. Moreover, annexin-V-propidium iodide staining was performed to confirm the degree of apoptosis-induced cell death. Both A375SM and A375P cells showed an increase in the early/late apoptosis rate after chrysin treatment. In a previous study, the rate of early/late apoptosis was found to increase when PC-3, a prostate cancer cell line, was treated with 40 µM of chrysin [36]. These results suggest that the decrease in the viability of the A375SM and A375P cells is caused by chrysin-induced apoptosis. Proteins that regulate apoptosis include the Bcl-2 family members, including Bid, Bax, and Bak, which induce apoptosis; and Bcl-xL, Bcl-2, and Mcl-1, which inhibit apoptosis [13]. These proteins (such as Bax and Bim) affect the intrinsic pathway and cause mitochondrial dysfunction to release cytochrome C, which induces caspase activity and apoptosis [37]. Therefore, the expression pattern of these proteins in both melanoma cells after chrysin treatment was confirmed. An increase in Bax and a decrease in Bcl-2 were confirmed in both cell lines. Fragmentation of PARP and caspase was observed. These results suggest that chrysin induces apoptosis in both melanoma cell lines through the intrinsic pathway involving the mitochondria. 

Autophagy is a physiological process that occurs within a cell. In the initial stage, a single membrane surrounds the cytoplasm and intracellular organelles, and a double membrane is formed later, forming structures called autophagosomes. Thereafter, the autophagosomes bind to lysosomes to form autolysosomes, and the isolated contents in the compartment are degraded by hydrolase [19]. Here, when the melanoma cells were treated with chrysin, vacuole formation was observed, in contrast to the control group. Acidic vesicular organelles increased after staining with acridine orange and both cell lines showed an increase in the protein expression of Beclin 1 and LC3 in the chrysin-treated cells. In a previous study, both Beclin 1 and LC3 were shown to increase when HEC-1A endometrial cancer cells were treated with chrysin [38]. These results suggest that chrysin similarly induces autophagy in A375SM and A375P cells. Generally, autophagy is involved in cell survival to protect cells; however, it also plays a specific role in cell death [20,21,22]. Therefore, a study on how autophagy and apoptosis are linked is considered important in the field of apoptosis [39]. Here, cell viability was confirmed following the pretreatment of both A375SM and A375P cells with 3-MA and HCQ. In a previous study, when quercetin suppressed late-stage autophagy using chloroquine in AGS and MKN28 gastric cancer cells, cell viability decreased further [40]. Similarly, here, in A375SM, 3-MA did not show a significant difference; however, HCQ treatment resulted in a greater decrease in cell viability than chrysin treatment alone. When silibinin inhibited early- and late-stage autophagy with 3-MA and bafilomycin A1 in MCF-7 breast cancer cells, both inhibitors showed a tendency to decrease cell viability [41]. In contrast, when chrysin-induced autophagy was suppressed using 3-MA in A375P, cell viability increased. However, there was no difference when HCQ was used. These results suggest that chrysin-induced autophagy has a protective role in A375SM cells; however, it induces apoptosis in A375P cells. Moreover, the effects are different depending on the cell type or the period of inhibition, even for the same substance. 

The mTOR pathway is involved in cell survival, including cell proliferation, growth, and differentiation. It can also activate processes such as tumor initiation and progression. Previous studies have reported that the mTOR pathway affects autophagy and apoptosis. Therefore, mTOR pathway modulation could be a potential target for anticancer therapy [27,28,29]. An increase in mTOR results in the phosphorylation of p70S6K (ribosomal protein S6 kinase) and 4EBP1 (4E binding protein 1). mTOR also initiates protein synthesis and elongation and increases the translation ability of mRNA [42]. In a previous study, isoquercitrin showed a tendency to induce apoptosis and autophagy by inhibiting the downstream factor p70S6K through the inhibition of mTOR in liver cancer cells [43]. Moreover, echinatin, another flavonoid, induced apoptosis and autophagy by inhibiting Akt and mTOR in esophageal cancer [44]. Similarly, the results of this study showed that chrysin inhibits the downstream factors p-p70S6K, p-S6K, and p-4EBP1 through the inhibition of p-mTOR in A375SM and A375P cells, and that it induces apoptosis and autophagy. 

In summary, the results of this study suggest that chrysin inhibits the viability of A375SM and A375P melanoma cells through the induction of apoptosis and autophagy. Chrysin-induced apoptosis might be instigated through the intrinsic pathway by increasing Bax and inhibiting Bcl-2. Moreover, the induction of autophagy was confirmed by the increase in Beclin 1 and LC-3. A decrease in cell viability was observed in A375SM after late-stage autophagy inhibition. In contrast, the viability of A375P cells increased after the early-stage inhibition of autophagy. These results suggest that, even with the same substance, the effect of autophagy inhibition differs depending on the cell type or the inhibition period. Moreover, chrysin can potentially be a naturally derived adjuvant for the treatment of skin cancer, as it inhibits mTOR/S6K. Additional studies on the molecular mechanisms related to the associations of chrysin-induced apoptosis and autophagy and efficacy-related experiments in vivo are necessary.

## Figures and Tables

**Figure 1 biomedicines-10-01467-f001:**
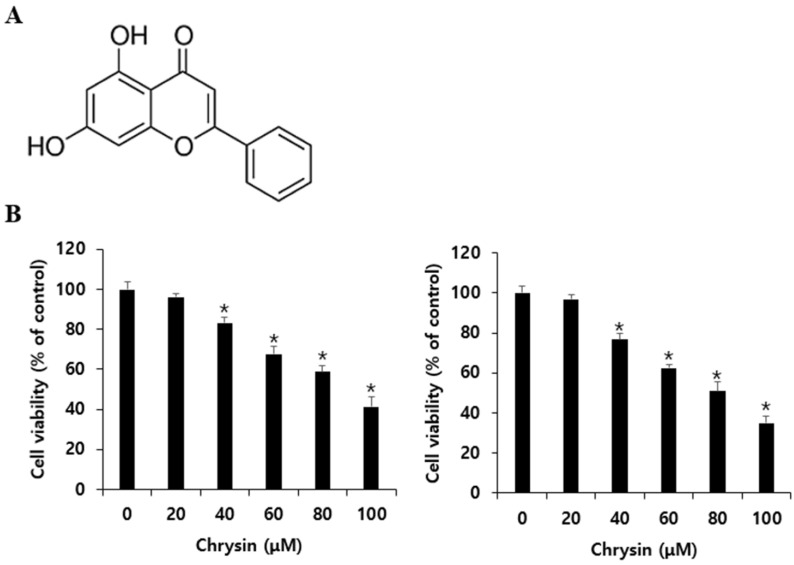
Chrysin inhibits cell viability in A375SM and A375P human melanoma cancer cells. (**A**) Chemical structure of chrysin. (**B**) A375SM and A375P cells were seeded in 96-well plates, incubated for 24 h, and then treated with the indicated concentration of chrysin for 24 h. Cell viability was measured by MTT assay. Data are presented as the mean and standard deviation for three samples. Significance was determined by Student’s *t*-test, * *p* < 0.05, compared with the control.

**Figure 2 biomedicines-10-01467-f002:**
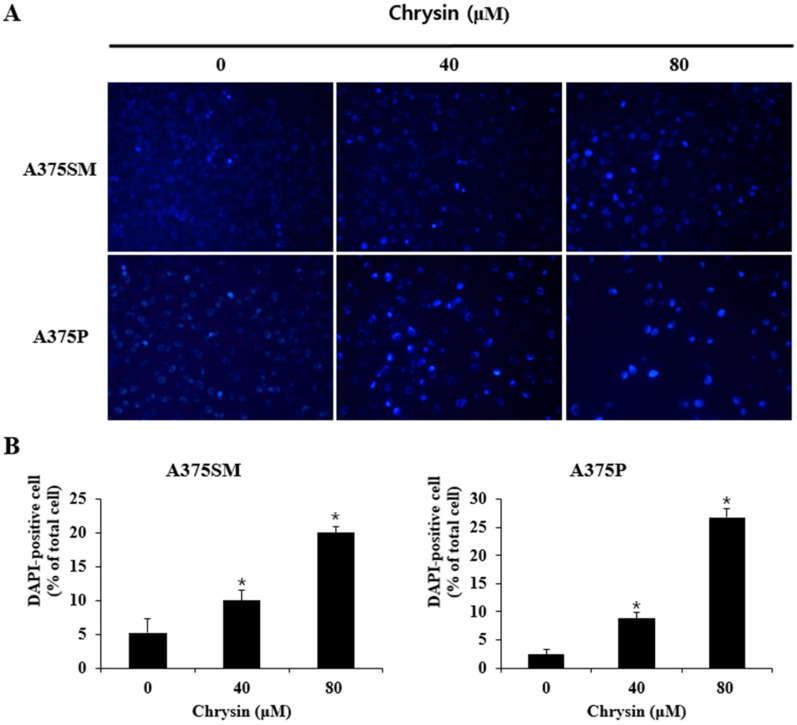
Chrysin induces apoptotic body formation in A375SM and A375P human melanoma cancer cells. (**A**) Cells were seeded for 24 h and then incubated with the indicated concentration of chrysin for 24 h. After that, cells were fixed and stained with DAPI solution. Cell morphological changes were analyzed using a fluorescence microscope (200×). (**B**) The bar graph represents the average of four fields under a fluorescence microscope and the percentage of DAPI-positive cells among all cells. Data are presented as the mean and standard deviation for three samples. Significance was determined by Student’s *t*-test, * *p* < 0.05, compared with the control.

**Figure 3 biomedicines-10-01467-f003:**
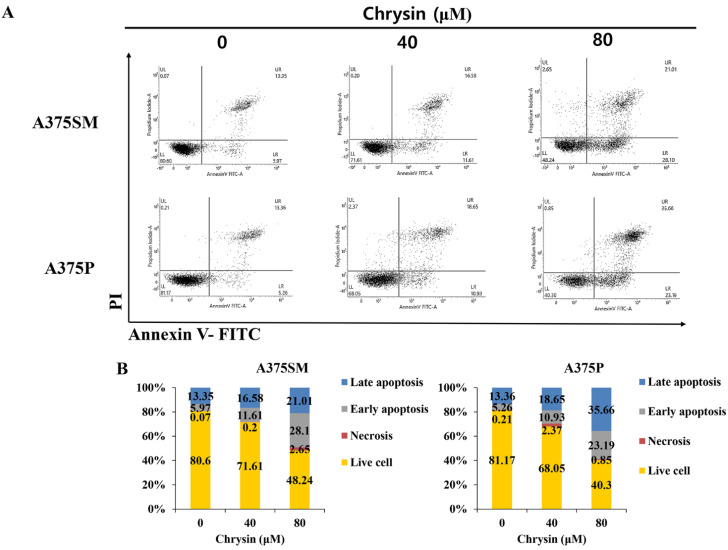
Chrysin induces apoptosis in A375SM and A375P human melanoma cancer cells. (**A**) Cells were seeded for 24 h. After treatment with the indicated concentrations of chrysin for 24 h, cells were harvested, stained with annexin V−propidium iodide, and analyzed by flow cytometry for apoptosis rates. (**B**) Graphs represent the percentages of living cells, necrotic cells, and early and late apoptotic cells.

**Figure 4 biomedicines-10-01467-f004:**
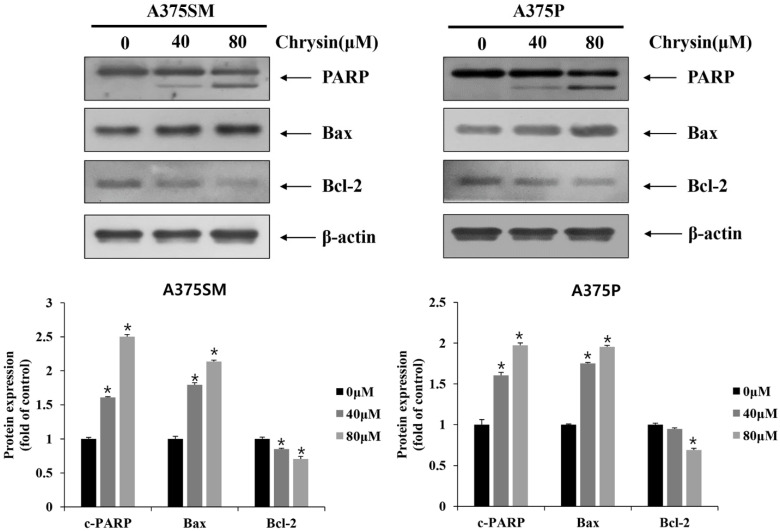
Chrysin induces apoptosis protein expression in A375SM and A375P human melanoma cancer cells. Cells were treated with the indicated concentrations of chrysin and the levels of poly ADP−ribose polymerase, Bax, and Bcl−2 proteins were measured by Western blotting. β−actin was used as a loading control. Quantification was performed in three independent experiments using ImageJ, normalizing the intensities of the bands to the controls. Data are presented as the mean and standard deviation for three samples. Significance was determined by Student’s *t*-test, * *p* < 0.05, compared with the control.

**Figure 5 biomedicines-10-01467-f005:**
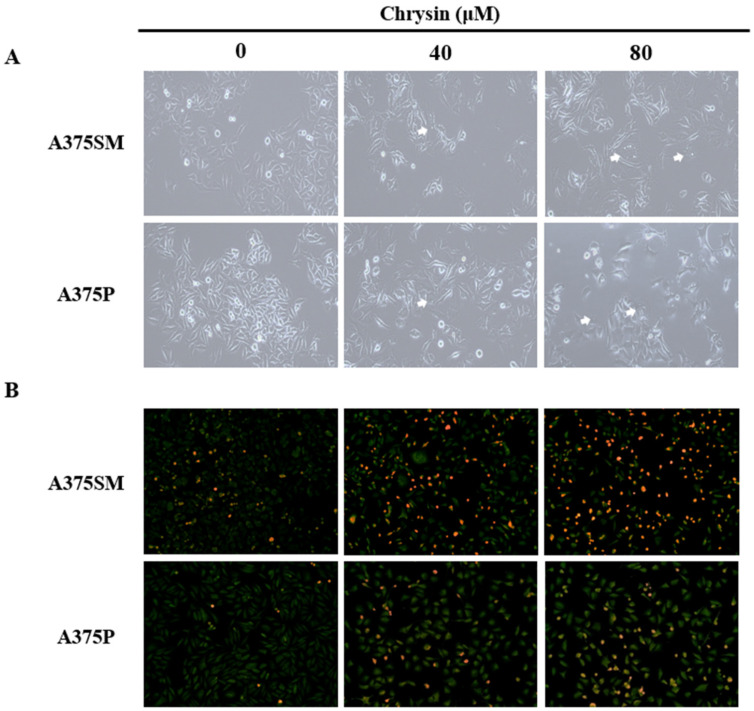
Chrysin induced autophagic vacuole formation in A375SM and A375P human melanoma cancer cells. Cells were treated with the indicated concentrations of chrysin for 24 h. (**A**) Morphological changes were observed under phase-contrast microscopy. The white arrow indicates the autophagic vacuoles. (**B**) Cells were stained with 5 µg/mL acridine orange to detect acid vesicle organelles and observed using a fluorescence microscope. The cytoplasm and nucleus fluoresce green and the acidic vesicular organelles fluoresce red.

**Figure 6 biomedicines-10-01467-f006:**
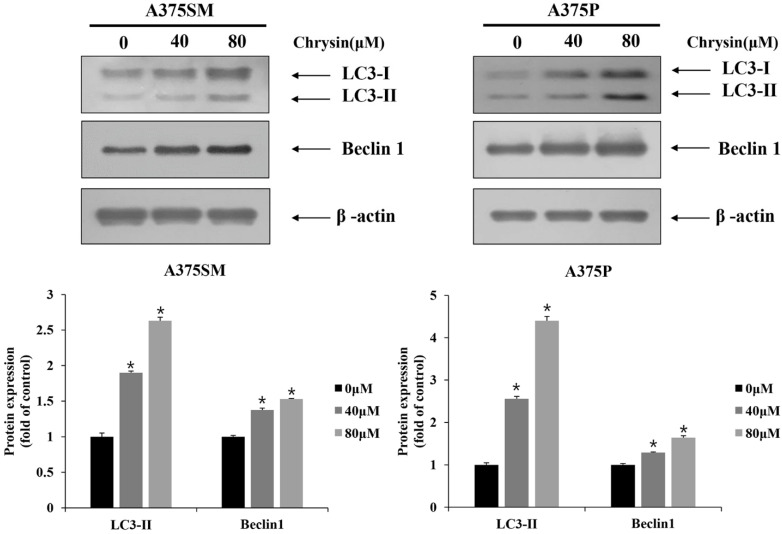
Chrysin induces autophagy proteins in A375SM and A375P human melanoma cancer cells. Cells were treated with the indicated concentrations of chrysin and protein levels of LC3 and Beclin 1 were measured by Western blotting. β-actin was used as a loading control. Quantification was performed in three independent experiments using ImageJ, normalizing the intensities of the bands to the controls. Data are presented as the mean and standard deviation for three samples. Significance was determined by Student’s *t*-test, * *p* < 0.05, compared with the control.

**Figure 7 biomedicines-10-01467-f007:**
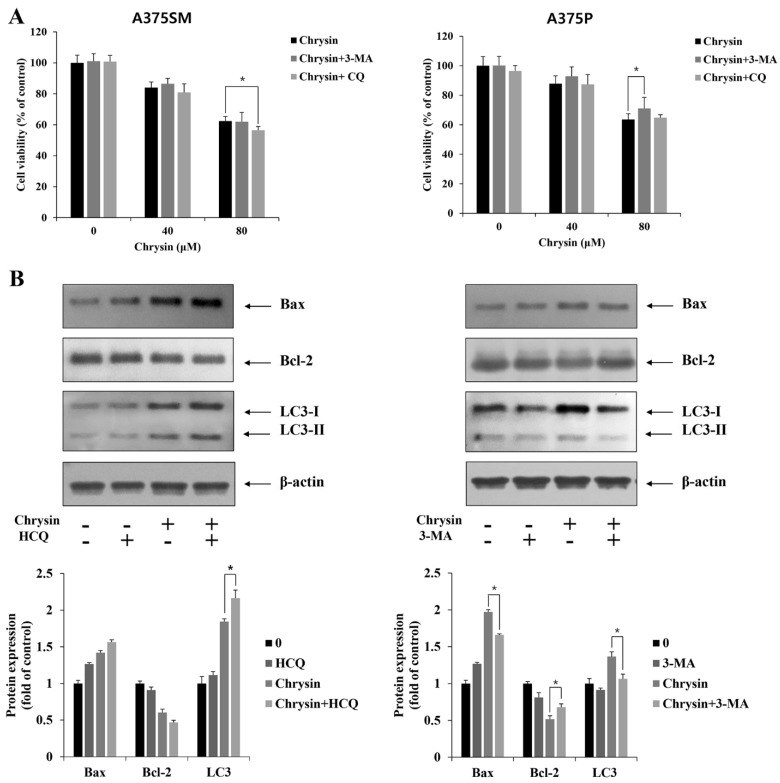
Chrysin induced cell death or cell-protective autophagy in A375SM and A375P human melanoma cancer cells. Cells were pretreated with hydroxychloroquine (HCQ, 25 μM, 2 h) or 3−methyladenine (3−MA, 2 mM, 1 h) and subsequently treated with indicated concentrations of chrysin for 24 h. (**A**) Cell viability was measured by MTT assay. (**B**) Protein levels of Bax, Bcl−2, and LC3 were measured by Western blotting. β−actin was used as a loading control. Quantification was performed in three independent experiments using ImageJ, normalizing the intensities of the bands to the controls. Data are presented as the mean and standard deviation for three samples. Significance was determined by Student’s *t*-test, * *p* < 0.05, compared with the control.

**Figure 8 biomedicines-10-01467-f008:**
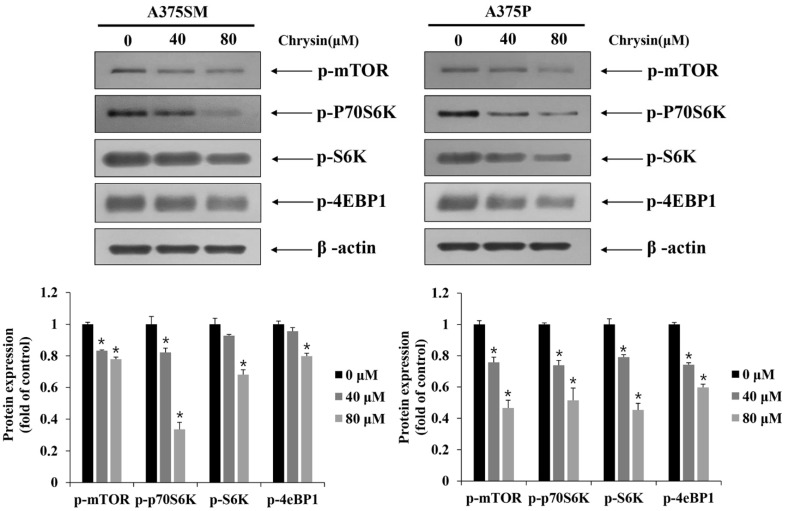
Chrysin inhibited the mTOR/S6K pathway in A375SM and A375P human melanoma cancer cells. Cells were treated with the indicated concentrations of chrysin and protein levels of p-mTOR, p-p70S6K, p-S6K, and p-4EBP1 were measured by Western blotting. β-actin was used as a loading control. Quantification was performed in three independent experiments using ImageJ, normalizing the intensities of the bands to the controls. Data are presented as the mean and standard deviation for three samples. Significance was determined by Student’s *t*-test, * *p* < 0.05, compared with the control.

## Data Availability

Not applicable.

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
