# Peer review of "Chrysin Induces Apoptosis and Autophagy in Human Melanoma Cells via the mTOR/S6K Pathway"

_biomedicines, 2022, doi:10.3390/biomedicines10071467_

Round 1

Reviewer 1 Report

The present manuscript is well organized and quite insightful. However, this manuscript needs some minor changes before its consideration for the publication.

Comments:

1. The whole manuscript is well organized except abstract of the manuscript. Reviewer suggests reconsidering the whole abstract and trying to write it in more meaningful and informative way.

2. Authors have used two concentrations of Chrysin (40 and 80 µM) in all the experiments for uniformity except in Acridine orange staining experiment where authors mentioned different concentrations (60 and 100 µM in material and methods). Reviewer could not understand the use of this variability in concentration of Chrysin. Moreover, in Figure 5 which corresponds to the experimental details of Acridine orange staining, author has mentioned Chrysin concentrations 40 and 80 µM. Thus, Reviewer suggests to either reconsider this experiment or provide relevant discussion of this.

3. There is no results and discussion for Figure 7. Moreover, Figure 8 results are not matching with its mentioned results. Reviewer suggests reconsidering the result section corresponding to both the figures (Figure 7 and 8).

Reviewer 2 Report

The authors report Chrysin induces apoptosis and autophagy in human melanoma 2 cells via the mTOR/S6K pathway, is well written and interesting. I am just concerned to know the below information.

Line 121: Authors selected 0, 40, and 80 µM concentrations for DAPI Staining whereas 0, 60, and 100 µM concentrations for Acridine Orange Staining and other tests, why, better to explain this in the discussion part?

Samples were treated with Chrysin for 24 h, How did the authors decide the duration? Provide some references for the same. 
